# Post-Transcriptional Control of Mating-Type Gene Expression during Gametogenesis in *Saccharomyces cerevisiae*

**DOI:** 10.3390/biom11081223

**Published:** 2021-08-17

**Authors:** Randi Yeager, G. Guy Bushkin, Emily Singer, Rui Fu, Benjamin Cooperman, Michael McMurray

**Affiliations:** 1Department of Cell and Developmental Biology, University of Colorado Anschutz Medical Campus, Aurora, CO 80045, USA; randi.yeager@cuanschutz.edu (R.Y.); esinger@skidmore.edu (E.S.); benjamin.j.cooperman@cuanschutz.edu (B.C.); 2Whitehead Institute for Biomedical Research, Cambridge, MA 02142, USA; guybushkin@gmail.com; 3RNA Biosciences Initiative, School of Medicine, University of Colorado, Aurora, CO 80045, USA; rui.fu@cuanschutz.edu

**Keywords:** budding yeast, sporulation, gametogenesis, gene expression, splicing, antisense

## Abstract

Gametogenesis in diploid cells of the budding yeast *Saccharomyces cerevisiae* produces four haploid meiotic products called spores. Spores are dormant until nutrients trigger germination, when they bud asexually or mate to return to the diploid state. Each sporulating diploid produces a mix of spores of two haploid mating types, **a** and α. In asexually dividing haploids, the mating types result from distinct, mutually exclusive gene expression programs responsible for production of mating pheromones and the receptors to sense them, all of which are silent in diploids. It was assumed that spores only transcribe haploid- and mating-type-specific genes upon germination. We find that dormant spores of each mating type harbor transcripts representing all these genes, with the exception of Mat**a**1, which we found to be enriched in **a** spores. Mat**a**1 transcripts, from a rare yeast gene with two introns, were mostly unspliced. If the retained introns reflect tethering to the *MAT***a** locus, this could provide a mechanism for biased inheritance. Translation of pheromones and receptors were repressed at least until germination. We find antisense transcripts to many mating genes that may be responsible. These findings add to the growing number of examples of post-transcriptional regulation of gene expression during gametogenesis.

## 1. Introduction

The establishment of mating type in the budding yeast *Saccharomyces cerevisiae* was one of the first genetic circuits controlling cell fate to be understood in mechanistic detail. The circuitry and signal transduction pathways by which mating types manifest have been reviewed extensively [1,2,3,4] (Figure 1a). Two mating types, **a** and α, differ genetically by only a few thousand basepairs of DNA at a single locus (*MAT*), representing two alternative alleles that encode either the Mat**a**1 and Mat**a**2 or the Matα1 and Matα2 proteins. Mat**a**2 has no known function. Mat**a**1 and Matα2 are homeodomain proteins that together form a heterodimeric transcriptional repressor of a small number of genes (~20) important for mating by haploid cells of both mating types [5], including MATα1. Mat**a**1–Matα2 also represses *RME1*, which inhibits meiosis and sporulation, the partially coupled processes by which diploid cells produce (usually) four haploid spores, two **a** and two α. Matα2 binds to the constitutively expressed protein Mcm1 to form a heterotetramer that represses a smaller number of genes (seven) mostly important for mating by **a** cells [5]. Finally, Matα1 and Mcm1 form a complex that activates five genes mostly important for mating by α cells [5]. **a** cells produce and secrete a pheromone (**a**-factor) that is detected by a G-protein-coupled receptor expressed only in α cells, and α cells produce and secrete a pheromone (α-factor) that is detected by a G-protein-coupled receptor expressed only in **a** cells. Thus the few gene expression differences arising from the identity of the allele at *MAT* are sufficient to confer three distinct cell fates: mating as **a**, mating as α, or non-mating but capable of meiosis/sporulation (**a**/α).

The genetic circuitry that controls these cell fates is also nimble. Haploid cells of most natural isolates of *S. cerevisiae* are able to switch mating types. Mating-type switching is thought to have evolved to allow isolated haploid spores a relatively rapid pathway to diploidy [8]. Switching occurs by a gene conversion event following a double-strand DNA break at *MAT*, swapping with the silent mating type loci cassettes harboring the *MAT***a***1* and *MAT***a***2* or the *MATα1* and *MATα2* protein-coding genes [4]. The phenotypic switch is completed within a cell cycle after the genetic switch [9], indicating that all proteins dictating sexual identity, including the mating pheromone secreted and the receptor that senses the other pheromone (Ste3 or Ste2), are rapidly turned over and replaced by their mating-type-appropriate counterparts. Failure to turn over key proteins such as Ste2 or MATα2 upon mating-type switching leads to defects in the manifestation of the new sexual identity [10]. Mating itself represents another transition in cell type—from mating-competent to non-mating—that must be accomplished efficiently: after haploid cells fuse, the expression and/or activity of pheromones and their receptors must be quickly extinguished to prevent autocrine signaling leading to cell cycle arrest [11].

Sporulation is the other step in yeast life cycle that requires a transition between sexual identities, from non-mating (**a**/α) to mating-competent (**a** and α). Considering the importance in other transitions of preventing inappropriate expression of mating-type-determination genes, we were surprised to notice, in the data from a transcriptome-wide study of gene expression during sporulation [6], that haploid- and mating-type specific transcripts began to appear long before individual spores become isolated from each other by the de novo formation of cell membranes and walls (Figure 1b). Indeed, many such transcripts, including Ste2 and Ste3, appear even prior to meiotic recombination (Figure 1b) [6]. If these transcripts are inherited by every spore regardless of which allele at *MAT* it inherits, then an **a** spore would inherit transcripts normally appropriate for expression in an α spore, and vice versa. Here, we investigate this phenomenon by sorting spores according to their *MAT* allele and profiling their transcriptomes. We find evidence of multiple mechanisms that control sexual expression in spores.

## 2. Materials and Methods

Diploid strain FW2444 (SK1 background *P_CUP1_-IME1::kanMX/P_CUP1_-IME1::kanMX*) [12] was transformed using the Zymo Research Frozen-EZ Yeast transformation II Kit (#T2001) with *Stu*I-cut pBC58 [7] to integrate the *P_GPD1_-mCherry* cassette at *MAT***a**. The diploid strain FY2740 (*MAT***a***/MATα his3∆1/his3∆1 leu2∆0/leu2∆0 lys2∆0/lys2∆0 ura3∆0/ura3∆0 RME1(ins-308A)*/*RME1(ins-308A) TAO3(E1493Q)/TAO3(E1493Q) MKT1(D30G)/MKT1(D30G)* [13] was similarly transformed with uncut pNeo-MagicFluor [14], which was a gift from Leonid Kruglyak (Addgene plasmid #58564; http://n2t.net/addgene:58564, accessed on 16 August 2020; RRID:Addgene_58564). G-418 selection for this plasmid was maintained in the rich medium growth prior to transfer to sporulation medium. GBy6 is in the same background as FY2740 but homozygous for mutations in the *IME4* catalytic site (*MAT***a***/MATα his3∆1/his3∆1 leu2∆0/leu2∆0 lys2∆0/lys2∆0 ura3∆0/ura3∆0 RME1(ins-308A)/RME1(ins-308A) TAO3(E1493Q)/TAO3(E1493Q) MKT1(D30G)/MKT1(D30G) ime4*-(*D348A,W351A)*/*ime4*-*(D348A,W351A)*) [15]. GBy222 is *MAT***a***/MATα his3∆1/his3∆1 leu2∆0/leu2∆0 lys2∆0/lys2∆0 ura3∆0/ura3∆0 TAO3(E1493Q)/TAO3(E1493Q) MKT1(D30G)/MKT1(D30G)* [15]. GBy225 is *MAT***a***/MATα his3∆1/his3∆1 leu2∆0/leu2∆0 lys2∆0/lys2∆0 ura3∆0/ura3∆0*, with the S288C alleles at *RME1*, *TAO3*, and *MKT1* [15].

To prepare cells for RT-PCR, cells were grown in YPD supplemented with 4% glucose for 25 h at 30 °C with shaking and diluted to an OD_600_ of 0.2 in BYTA medium (1% yeast extract, 2% tryptone, 1% potassium acetate, 50 mM potassium phthalate). BYTA cultures were grown for an additional 16.5 h at 30 °C with shaking. Cells were washed once with water, re-suspended in 0.3% potassium acetate to OD_600_ of 2, and incubated at 30 °C with shaking.

To prepare spores for sorting by mating type, 25-mL cultures of FW2444 cells with integrated pBC58 were grown in liquid YPD (2% peptone, 2% glucose, 1% yeast extract) medium supplemented with (per L) 0.096 g L-tryptophan, 0.024 g uracil, and 0.012 g adenine at 30 °C in 250-mL flasks with shaking to exponential phase. Cells from these cultures were used to inoculate new 25-mL cultures of reduced-glucose YPD (1% glucose) supplemented with (per L) 0.024 g uracil and 0.012 g adenine to an OD_600_ of 0.05. These cultures were grown at 30 °C with shaking to OD_600_ of ~12, then pelleted and washed with sterile water twice, and suspended in copper-containing sporulation medium (1% potassium acetate, pH 7.0, 0.04 g/L adenine, 0.04 g/L uracil, 0.02 g/L histidine, 0.02 g/L leucine, 0.02 g/L tryptophan, and 0.02% raffinose, 50 µM copper (II) sulfate) to an OD_600_ of 2.5. Cells were incubated with shaking at 30 °C and harvested for sorting seven days after copper induction of sporulation.

For fluorescence-activated cell sorting (FACS), asci were digested enzymatically to remove the ascus wall and sorted essentially as described previously [7] using a BD FACSAria (BD Biosciences) in the Allergy and Clinical Immunology/Infectious Disease (ACI/ID) Flow Core of the University of Colorado Anschutz Medical Campus. Between 8 and 30 million cells of each color were used for RNA extraction.

### 2.1. RNA Sequencing and Data Analysis

Sorted spores were harvested by centrifugation and frozen in liquid nitrogen. Total RNA was isolated by hot acid phenol extraction, as follows. Thawed cells were resuspended in 400 µL cold TES (10 mM Tris-HCl, pH 7.5, 1 mM EDTA, 0.5% SDS). Then, 400 µL acid phenol:chloroform was added and the mixture was vortexed for 15 s in tubes pre-filled with garnet and zirconium satellites (#PFMM 500-100-25, OPS Diagnostics). The 15-s intervals of vortexing continued over the course of 30 min with 5-min intervals of incubation at 65 °C in between. After incubation on ice for 5 min, the tubes were centrifuged at 14,000 rpm for 10 min at 4 °C. The aqueous layer was transferred to a new tube and the RNA was precipitated with sodium acetate and ethanol. Then, 100 ng total RNA was used as input for library preparation with a Tecan Universal Plus mRNA-seq Library Preparation Kit with NuQuant (Cat # 0520). Sequencing was completed using an Illumina NovaSEQ 6000 Instrument using v1.5 Chemistry with S4 Flow Cell at the Genomics Core of the University of Colorado Anschutz Medical Campus. Reads were aligned by STAR v2.7.3a to a modified version of the S288c reference genome (R64-1-1) containing sequences of the mCherry and hygromycin resistance gene, and then quantified by FeatureCounts from Subread using paired setting. Downstream analysis was performed in R using edgeR [16] to normalize data to counts per million for quantification of differentially expressed genes between mating types with a generalized linear model accounting for batch effect. To prepare images of read coverage, images were exported either from a custom track on the genome browser of the University of California Santa Cruz Genomics Institute [17] or using IGV (Integrative Genomics Viewer) version 2.4.19 [18].

### 2.2. Quantitative Reverse Transcription PCR

RNA was extracted from frozen cells (equivalent to 24 mL of culture at OD_600_ = 1) by first vortexing in 600 μL AE buffer (50 mM sodium acetate, 10 mM EDTA, 1% SDS) and 600 μL acid phenol in the presence of ~100 μL acid-washed glass beads at 4 °C for 5 min. The mixture was incubated at 65 °C for 10 min. Next, cells were vortexed again, incubated at 65 °C for 10 min again, vortexed, and pelleted at 18,400× *g* for 10 min at 4 °C. The aqueous top phase was transferred to a new tube and extracted again in phenol. After another centrifugation and transfer to a new tube, RNA was extracted in 400 μL chloroform followed by ethanol precipitation. cDNA was generated with SuperScript III (Life Technologies, Carlsbad, CA, USA) using random hexamers or gene-specific primers from 1 μg of total RNA. RT-qPCR was performed using SYBR green PCR master mix (Life Technologies, Carlsbad, CA, USA) with primers TTCCATCCAAGCCGTTTTGT and CAGCGTAAATTGGAACGACGT for *ACT1* and ATGGATGATATTTGTAGTATGGCG and TTATTTAGATCTCATACGTTTATTTATGA for all forms of unspliced and spliced Mat**a**1, or GAAATCAATCTCAATACTAATAATCTTT and ACGTTTATTTATGAATCTTACTTGAAGTG for *MAT***a***1* on Applied Biosystem 7500 or QuantStudio 5 instruments.

### 2.3. Microscopy

Cells were grown to saturated in liquid YPD (2% glucose) medium overnight at 30 °C, from which 200 µL was added to 5 mL of sterile water and pelleted. The cells were resuspended in 2.5 mL 1% potassium acetate and cultured at 30 °C, rotating in glass culture tubes for 4 days. To induce germination, aliquots of the sporulation culture were pelleted and resuspended in YPD (2% glucose) for 2 or 5 h, at which point they were pelleted again and resuspended in water before spotting a 3-µL aliquot on a 1% agarose pad on a microscope slide. After the liquid was absorbed by the pad, the cells were covered with a coverslip. Imaging was performed on an EVOSfl all-in-one microscope (Thermo-Fisher, Waltham, MA, USA) with a 60× oil objective and transmitted light, GFP, or Texas Red LED cubes. Images were processed with FIJI [19].

## 3. Results

### 3.1. Only One Transcript Distinguishes the Transcriptomes of Spores of Distinct Mating Types

Several “spore-autonomous” genes, including *GPD1* and *YKL050C*, have been previously exploited to create fluorescent markers that allow fluorescence-activated cell sorting (FACS) of spores carrying a specific allele at a given locus [20,21]. To separate **a** and α spores we used the FASTER-MT system, which relies on a cassette integrated into the *MAT***a***2* gene in which the *GPD1* promoter drives spore-autonomous expression of mCherry [7] (Figure 1c). We then isolated total RNA from the spore populations and sequenced mRNA using paired-end Illumina sequencing. We repeated the sporulation and sorting steps to generate two independent biological replicates. Spores exhibit little ongoing transcription [22,23]. Hence, we interpreted the presence of transcripts in spores as those produced and stably inherited during sporulation.

The abundance of only four transcripts was different between the two mating types, according to a false-discovery statistic of 5% (Figure 2a). Though the presence of mCherry protein was no guarantee of mRNA persistence, we found mCherry transcripts only in **a** spores, providing a positive control. We also found spore-autonomous inheritance of the transcript encoding the hygromycin resistance protein (hygromycin B phosphotransferase, or hph) that was used to select for integration of the *P_GPD1_-mCherry* cassette (Figure 2a,b). The heterologous promoter controlling expression of this gene, *TEF*, from the *Ashbya gossypii* gene encoding EF-1α, is expected to be constitutive during vegetative proliferation [24] but its activity during sporulation is unknown. If, like *YKL050C*, it is mainly active late in sporulation, after spore membranes close (Figure 1b), and if transcripts made earlier in sporulation are turned over quickly, this could explain the observed spore-autonomous inheritance. RNA isolation took place more than a week after sporulation was induced (see Section 2). Robust detection of mCherry and hph thus suggests that in mature spores these transcripts are extremely stable.

Our unbiased quantitative comparisons indicated that RNA-seq reads mapping to *YAR064W*, a fragment of a pseudogene located in a largely silent subtelomeric region, were also enriched in **a** spores (Figure 2a). However, closer inspection revealed that most reads mapped to the periphery of the *YAR064W* coding region and likely represented the ends of transcripts from adjacent, unannotated regions that themselves were not differentially represented (Figure 2b). Thus, we think there are no actual differences in expression or inheritance of *YAR064W*.

The final gene that was found in unequal abundance in spores of different mating types was Mat**a**1, which was enriched in **a** spores compared to α spores (Figure 2). Because the *P_GPD1_–mCherry* cassette is integrated nearby *MAT***a***1* (Figure 1c), we cannot exclude the possibility that whatever property of the *GPD1* promoter makes it spore-autonomous is conferred onto *MAT***a***1*. However, we find this explanation unlikely for two reasons. First, since the cassette is inserted between the first and second codons of Mat**a**2, the bidirectional *MAT***a***1/a2* promoter remains intact, and 296 basepairs separate the *P_GPD1_–mCherry* cassette from the *MAT***a***1* coding sequence. Second, the *GPD1* promoter only drives transcription of heterologous genes in one direction [25].

### 3.2. Accumulation of Unspliced Mata1 Transcripts in MATa Spores

Mat**a**1 harbors two small introns [26]. We noticed that the Mat**a**1 transcripts in **a** spores were incompletely spliced (Figure 2b). To independently confirm intron retention in Mat**a**1 transcripts, we harvested RNA at specific time-points from synchronously sporulating diploid cells and performed RT-PCR with Mat**a**1-specific primers, or primers targeting a control transcript, Act1, which encodes yeast actin. These data also showed intron retention in Mat**a**1 transcripts, particularly at later stages of sporulation (Figure 3a), consistent with published microarray-based studies [27] (Figure 3b). The appearance of “new”, unspliced transcripts during sporulation demonstrates that *MAT***a***1* transcription is ongoing, as opposed to a burst of early transcription in sporulating cells that produces extremely stable transcripts. Late in sporulation, the induction of ribosomal protein genes, ~100 of which contain introns, overwhelms the splicing machinery, leading to a widespread defect in splicing efficiency [27]. We chose 90 intron-containing genes somewhat arbitrarily (86 that were examined in a previous study [28] plus four randomly-selected ribosomal protein genes) and looked for evidence of intron retention in our spore RNA-seq data. Notably, 49 of the 90 genes (54%) had apparent intron retention, 23 were efficiently spliced and 18 were ambiguous (Table 1; see examples in Figure 3c). It has been proposed that the very short sizes of the Mat**a**1 introns (52 and 54 nucleotides) leads to inefficient splicing even in wild-type vegetative cells [29], but there was no clear bias toward short introns among those with retained introns: the minimum intron length was 56 or 58 nucleotides in the genes with or without intron retention, and the median and mean intron lengths were greater in the genes with retained introns than in those without (99 vs. 96 and 184.8 vs. 173.1, respectively). These findings suggest that introns are retained in many transcripts produced during sporulation. For Mat**a**1, the incomplete splicing was also associated with mating-type-specific RNA inheritance. Since we sorted spores solely on the basis of genotype at *MAT*, and our strain was homozygous at all other loci, our experiments do not address whether for other genes intron retention might correlate with biased segregation/inheritance during sporulation.

### 3.3. Antisense Transcripts Corresponding to “Mating-Type-Specific” Genes Are Pervasive in Spores

Our results demonstrate that the vast majority of genes present in a mating-type-specific manner in vegetative haploid cells are not mating-type-specific in spores, consistent with their accumulation early in sporulation before the cytoplasm is partitioned into the individual spores, and slow or no turnover thereafter. The same study reporting transcript levels during a time-course of sporulation also analyzed translation via ribosome profiling/Ribo-seq [6]. The numbers of ribosome-protected reads relative to the number of total reads were used to estimate translational efficiency. Ste3 had the 18th-lowest translational efficiency in unsorted spores [6]. Antisense transcripts corresponding to genes under mating-type control are known to affect both the ability to enter sporulation [30] and its progression [31]. We thus wondered whether antisense transcripts capable of regulating the translation of Ste3 might be made during sporulation. A large region of *STE3* overlaps with *YKL177W*, a dubious ORF with no known function transcribed in the opposite direction (Figure 4a). We noticed from published microarray studies that in vegetative cells differing only by mating type Ykl177w transcript levels correlate inversely with Ste3 levels (Figure 4b) [32]. By contrast, in spores of each mating type we found reads representing both Ykl177w and Ste3 transcripts (Figure 4a). We did not detect the putative antisense transcript MUT940 [31], which is complementary to the *STE3* 3′UTR (Figure 4a).

Examination of the published RNA-seq time-course of sporulation revealed that Ykl177w accumulates with kinetics similar to that of Ste3 (Figure 4c). The slightly earlier apparent induction of *YKL177W* compared to *STE3* likely avoids extensive RNA polymerase collisions that might result from simultaneous transcription. If these transcripts anneal to each other, the resulting dsRNA may inhibit Ste3 translation (see Section 4).

Our findings with *STE3* inspired us to look for antisense transcripts to other “mating-type-specific” genes. We examined 27 such genes as identified previously [5] and found evidence of antisense transcripts for 10 of them, *AFB1*, *AGA2*, *ASG7*, *BAR1*, *DDR2*, *FAR1*, *HO*, *MFA2*, *MFα2*, and *SAG1* (Figure 5 and Appendix A). With the exception of *DDR2*, we also found antisense transcripts for each of these 10 genes in published spore RNA-seq data [6]. We arbitrarily selected 10 control genes by adding or subtracting 100 from the systematic locus identifier for each of the 10 “antisense” genes. For example, for *HO* (=*YDL227C*), the corresponding control was *PCL2* (=*YDL127W*). For none of the control genes (*FRE7*, *IKS1*, *LAP2*, *NUP159*, *PCL2*, *RPS26A*, *RRN5*, *SNT2*, *SOD1*, *YJL070C*) did we find any evidence of antisense transcripts in spores (Appendix A and data not shown). These findings implicate the production of antisense transcripts during sporulation in the regulation of expression of “mating-type-specific” genes inherited by spores.

### 3.4. Translational Repression in Spores Mediated by the Promoters of Mating-Type-Specific Genes

The promoters of mating-type-specific genes also appear to be sufficient to confer translational repression in spores. In vegetative haploid cells carrying a plasmid with mCherry under control of the *STE3* promoter and GFP under control of the *STE2* promoter, the GFP is expressed only in **a** cells and mCherry only in α vegetative cells, as expected, but mature spores express neither reporter [14]. In spores fluorescence is observed only upon germination [14], a result we recapitulated (Figure 4d). Hence, unless in the plasmid context these promoters behave differently than the endogenous ones, the *STE2* and *STE3* promoters are each sufficient to repress translation of the transcripts they produce. Similarly, MFα1, encoding α-factor, was equally abundant in both spore populations (Figure 2a), yet in a published study a reporter in which GFP replaces the MFα1 ORF, GFP was detected only following germination and only in ~50% of spores (presumably *MATα*) [33]. Another way to estimate translational efficiency is to plot RNA reads against ribosome footprint reads. We plotted these published data for the mature spore samples obtained by the traditional synchronization sporulation method and noticed that *MFα*1 and three other mating-type-specific genes, *BAR1*, *SST2*, and *STE2*, were among the transcripts with the lowest number of ribosome-protected reads in the published data [6] (Appendix A). In summary, despite being inherited similarly by spores of each mating type, transcripts that are “mating-type-specific” in vegetative cells are particularly translationally repressed in spores. For those that have been tested directly (*STE2*, *STE3*, and *MFα1*), each displayed mating-type-specific expression only in germinating spores, and the promoters of these genes were sufficient to confer this regulation. While the underlying mechanisms in this context remain to be determined, the control of translation by yeast promoters is known to occur in other contexts [34].

## 4. Discussion

When it comes to meiosis/sporulation in *S. cerevisiae*, post-transcriptional control of gene expression appears to be the rule rather than the exception. There are numerous examples of genes that are transcribed early in the process but the protein products only appear later. Multiple mechanisms are at play and any could affect mating-type transcripts. While most examples of regulation by antisense transcripts in *S. cerevisiae* involve the inhibition of transcription in *cis*, in the case of “messenger-interacting mRNAs” (mimRNAs) produced by convergently transcribed genes, ribosomes stall at the site of overlap [35]. We detected complementary RNAs in the same cells which, if they form dsRNA, could interfere with translation. The only gene identified in both screens [36,37] for mutants specifically defective in germination, *TRF4/PAP2*, encodes a subunit of the TRAMP (Trf4/5-Air1/2-Mtr4-Polyadenylation) complex involved in degrading “aberrant” nuclear RNAs [38,39,40]. We speculate that upon germination the TRAMP complex targets incompletely processed nuclear transcripts, potentially including dsRNAs resulting from antisense–sense annealing, and that mating-type-specific genes are enriched among this class of transcripts in spores. TRAMP eliminates “aberrant” nuclear transcripts by polyadenylating them in a way that targets them for degradation by the exosome [39]. Interestingly, in the fission yeast *Schizosaccharomyces pombe*, genes that function in meiosis are transcribed in vegetative cells but the nascent transcripts are bound by the Mmi1 protein in a sequence-dependent manner, which inhibits their splicing and promotes hyperadenylation and consequent destruction by the exosome [41,42,43]. There is no clear Mmi1 counterpart in *S. cerevisiae*.

This model is consistent with the fact that spore germination is known to drive rapid and extensive decay of mRNAs that are otherwise stable for months [22], though decay could also follow a burst of translation. Microarray experiments showed that the relative abundance of mating transcripts increases starting about an hour after germination and peaking after about two hours, even in isolated spores not exposed to pheromone [23]. Thus the germination-specific expression of reporter genes under the control of *STE2*, *STE3*, or the *MFα1* promoter may reflect translation of transcripts made following germination. Interestingly, mating transcripts also accumulate in spores exposed only to glucose, but such spores are unable to mate, pointing to persistent translational inhibition only relieved by some additional process requiring other signals (e.g., a nitrogen source) to induce the full germination program [23]. One caveat to those experiments is that relative increases in the abundance of specific transcripts during germination could represent either new transcription or persistence of those transcripts and decay of most others.

In addition to the appearance of antisense transcripts and differential splicing, during sporulation there are widespread changes in usage of alternative transcription and translation start sites (Kim Guisbert et al. 2012; Cheng et al. 2018; Eisenberg et al. 2020). Our current data do not address these mechanisms for mating-type-specific genes in spores. Modification of *S. cerevisiae* RNAs by N^6^-methyladenosine (m^6^A) occurs only during sporulation, via the methyltransferase Ime4 [44,45]. >1100 transcripts, including Ste3 and Ste6, are m^6^A-modified but functional roles are mostly unknown [45]. Ime4 is nuclear, and m^6^A modification of nascent transcripts could inhibit RNA processing and nuclear export, or could alter mRNA half-life. Mat**a**1 does not appear to be m^6^A modified [15]. Thus, we think it is unlikely that m^6^A directly affects Mat**a**1 splicing or stability.

RNA-binding proteins bind and control the translational timing of various mRNAs that are transcribed early during sporulation but are not translated until later [46,47,48]. Whether mating-type transcripts are bound in messenger ribonucleoprotein (mRNP) complexes during sporulation and in mature spores is unknown. However, a recent study showed that in vegetative cells many mating-type RNAs, including Ste3 and the pheromone-encoding transcripts MFα1 and MFα2, are bound together in mRNPs that the authors called “transperons” [49]. Transcripts in transperons co-localize both before and after nuclear export and were presumed to have similar translational fates [49]. The *STE2* and *AGA2* loci come in close proximity in vegetative cells, suggesting that transperon assembly is co-transcriptional [49]. A *cis*-acting 47-nucleotide sequence found in *STE2* and four other mating genes is important for transperon assembly [49]. The fact that these sequences, like the antisense Ste3 transcripts we found in spores, are in coding regions suggests that neither transperon- or antisense-mediated repression is sufficient to explain the promoter-driven translational repression we observed for *STE2* and *STE3* using the reporter plasmid (Figure 4d).

We were surprised to find that the transcriptomes of spores of both mating types were so similar. At first glance, the presence of normally **a**-specific transcripts in an α spore would seem to risk sexual confusion upon germination. Indeed, ectopically expressed Ste3 in **a** cells does bind the **a**-factor produced by the same cells, but this autocrine signal does not trigger the downstream responses that normally accompany pheromone sensing. Instead, the activity of the G protein β subunit, Ste4, is inhibited in a process termed receptor inhibition, blocking downstream signaling and thus inhibiting the ability of α-factor-bound Ste2 to initiate the mating response [11,50,51,52]. When spores germinate they frequently avail themselves of a nearby mating partner and mate immediately following a short period of outgrowth of new cell wall [33,53]. In this context, receptor inhibition resulting from the presence of both pheromones and both receptors would disfavor mating and promote budding. In fact, in some *S. cerevisiae* strains, this is precisely what is observed: spores forego sex and bud instead [33]. For strains in which spores were more likely to mate, α-factor production was generally higher and, for some, the MFα1 gene encoded more copies of the pheromone-peptide-encoding sequences [33]. Some degree of receptor inhibition due to translation of “residual” inappropriate transcripts like Ste3 could contribute to this state of limiting pheromone. Interestingly, the transcript with the lowest composite translational efficiency in spores is Dig2 [6]. Dig2 protein binds Ste12, a key activator of mating genes in both mating types, and both inhibits it and protects it from degradation. The resulting pool of inactive Ste12 becomes active when pheromone triggers Dig2 phosphorylation and release from Ste12 [54,55,56,57,58,59]. The extent to which translational repression of inherited Dig2 transcripts is reversed upon spore germination could also influence the bud-vs-mate decision. Overall, then, the simplest model for the fate of inherited transcripts representing a mix of both mating types is that they are mostly translationally repressed and have minor, if any, phenotypic consequences.

Though we cannot rule out an unknown function in spores, Mat**a**1 protein has no known function in vegetative **a** cells. If Mat**a**1 protein is present in cells that also express MATα2, the Mat**a**1–Matα2 heterodimer, which is rather long-lived [60], might repress transcription of important mating genes. Hence from the perspective of spore mating the most important fate of Mat**a**1 is its absence in MATα cells. The accumulation of unspliced Mat**a**1 late in sporulation offers two possible mechanisms of biased inheritance. Slow Mat**a**1 splicing means that fully spliced Mat**a**1 was probably transcribed early. In vegetative cells unspliced Mat**a**1 transcript escapes cytoplasmic nonsense-mediated decay and is instead destroyed by nuclear RNA decay pathways, which has been proposed to reflect nuclear retention of the unspliced transcript [29]. The decrease in fully spliced Mat**a**1 late thus may indicate RNA decay from which unspliced or newly-made Mat**a**1 is protected. Decay of spliced Mat**a**1 past the point of meiotic anaphase II and spore isolation would enforce spore autonomy of Mat**a**1 inheritance. A second, non-exclusive mechanism relates to the fact that in RNA processing mutants some nascent transcripts remain associated with the locus of origin [61]. Thus, Mat**a**1 transcripts could be physically linked to the *MAT***a***1* gene and segregate with it during meiotic chromosome segregation.

Indeed, in mammalian cells, there are multiple examples of programmed intron retention as an apparent cellular strategy to sequester transcripts in the nucleus and even on chromatin, preventing the synthesis of proteins in inappropriate developmental contexts [62,63]. In meiotic spermatocytes, intron retention—resulting from high transcription rates outpacing splicing capacity—leads to long-lived transcripts encoding proteins with key functions in gametes [62]. While parallels with our observations are striking, a crucial distinction is that, rather than males producing sperm and females producing eggs, in *S. cerevisiae* a single cell type (**a**/α) produces gametes of two distinct mating types, necessitating an additional layer of regulation. Dissecting in further detail the genetic circuitry underlying this example of cellular differentiation will be fertile ground for future research.

## Figures and Tables

**Figure 1 biomolecules-11-01223-f001:**
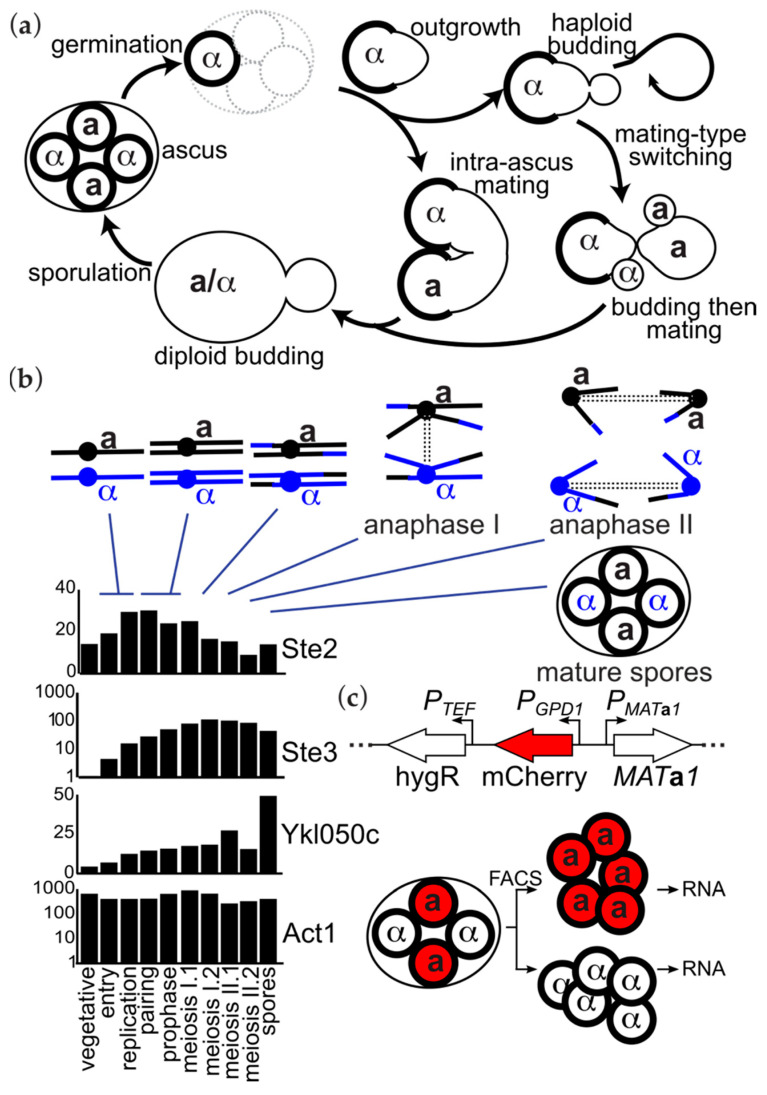
Early accumulation of mating-type-specific transcripts during *S. cerevisiae* and an approach to determine their inheritance in spores separated by mating type. (**a**) A simplified illustration of the life cycle of *S. cerevisiae*. Diploid cells undergo meiosis and sporulation, typically producing four haploid spores within each sporulating cell. Each spore is encased in a specialized wall (thicker lines) that confers resistance to a variety of environmental stressors. The two mating types, **a** and α, reflect alternative alleles at a single locus. Meiosis usually produces two pairs of spores of opposite mating types, which remain largely dormant until they germinate. After germination, spores can either bud or mate, often with another spore from the same ascus. Haploid budding can proceed indefinitely, or mating can occur at any point, given an appropriate partner. A haploid spore from most natural isolates is able to switch mating types and, via subsequent mating with one of its offspring, return to the diploid state. (**b**) Bar graphs show published data [6] with levels of the indicated transcripts as assessed by RNA-seq from cells taken at various time-points from synchronously-sporulating cultures. Major cellular events at each time-point are listed below and illustrated above, for a single representative chromosome (Chr. III) that also harbors the mating-type locus near its centromere. Data represent composite values from two methods of synchronized sporulation. Note that some *y* axes are in log scale. The spore-autonomous gene *YKL050C* is shown as an example of the expression pattern expected for transcripts inherited only by the spores that inherit the encoding allele. (**c**) Illustration of the FASTER-MT method [7] for isolating **a** and α spores via fluorescence-activated cell sorting (FACS). The spore-autonomous *GPD1* promoter drives expression of mCherry and is inserted at the mating-type locus with the **a** allele, encoding Mat**a**1. “hygR”, the hygromycin-resistance gene *hph* under control of the heterologous *TEF* promoter. Illustration is not to scale.

**Figure 2 biomolecules-11-01223-f002:**
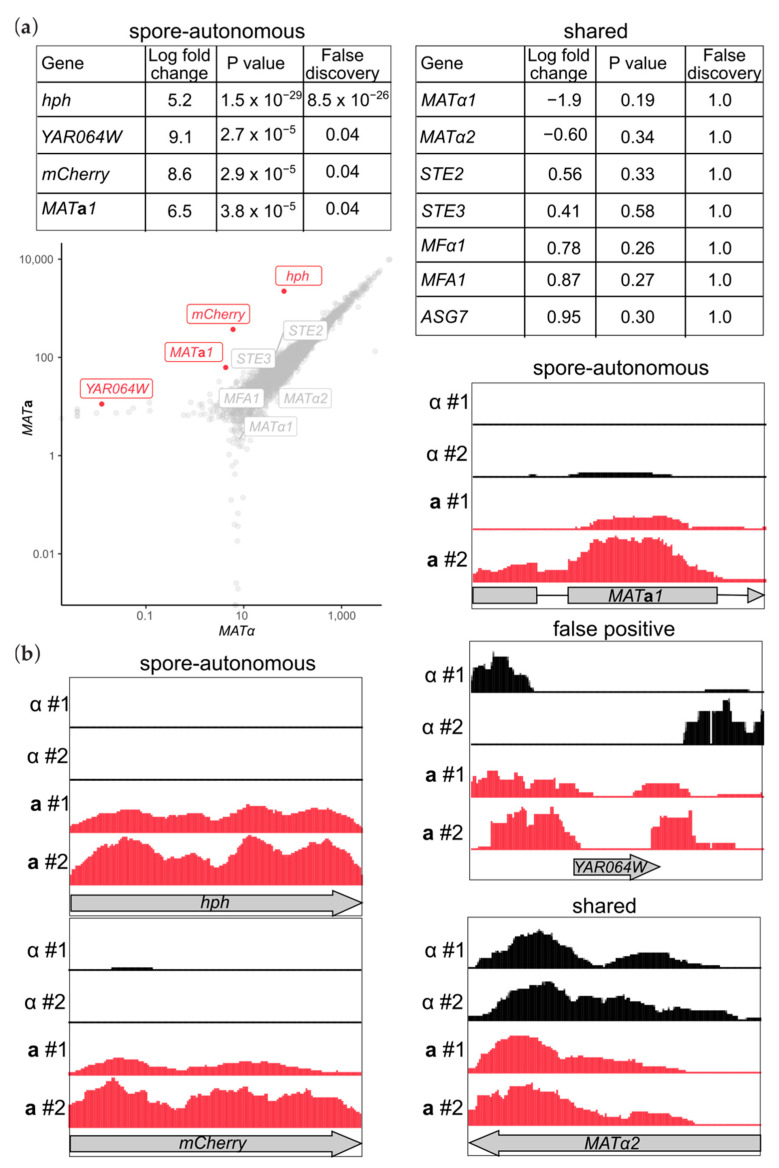
Inheritance of most “mating-type-specific” transcripts is independent of spore mating type. (**a**) RNA-seq data were analyzed by edgeR to identify transcripts for which abundance differed significantly between spores of each mating type. Values for ten transcripts of interest are shown in the tables. All but the four in the “spore-autonomous” table were shared approximately equally, according to a false-discovery statistic of <0.05. The correlation plot below highlights those four in red. (**b**) RNA-seq reads mapped onto the genome for the two biological replicates of the red (“**a**”) and not red (“α”) sorted spores. Shown are the four transcripts considered “spore-autonomous” in (**a**) and Matα2 as an example of a “shared” transcript. Gene structures are illustrated below. For each transcript, the scale of the *y* axis is the same between all four samples.

**Figure 3 biomolecules-11-01223-f003:**
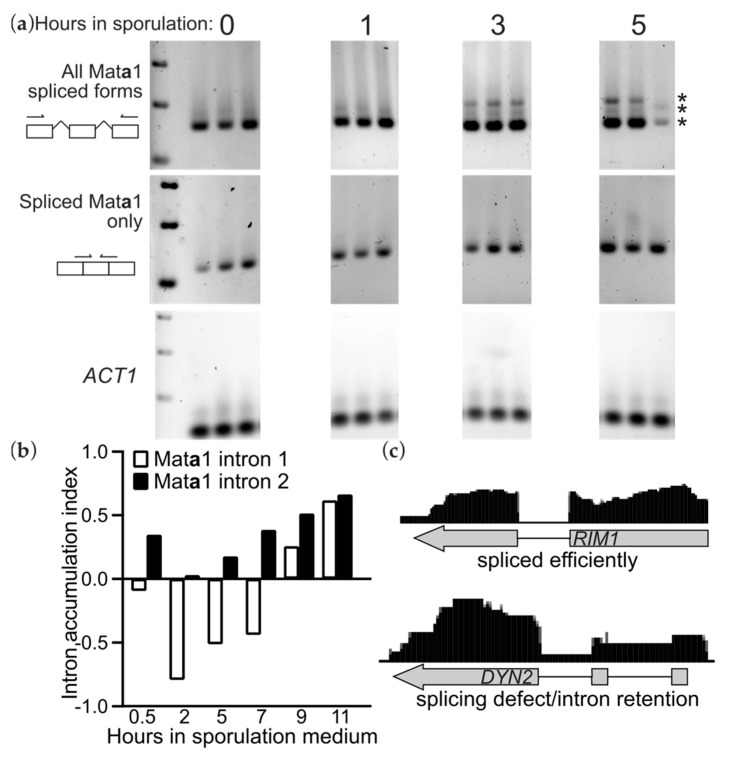
Intron retention in Mat**a**1 transcripts reflects a widespread splicing defect. (**a**) RT-PCR was used to detect Mat**a**1 transcripts in total RNA isolated from cells of strain FY2740 undergoing synchronous sporulation. Three biological replicates were analyzed for each sample. The left-most lane is a molecular weight marker (SM0311, Thermo-Fisher, Waltham, MA, USA) showing bands of 750, 500, and 250 basepairs. Illustrations to the left show where the Mat**a**1 primers anneal and how they allow detection of all spliced forms or fully-spliced Mat**a**1. Asterisks at right mark the three spliced forms. (**b**) Published microarray data [27] showing Mat**a**1 intron retention in cells undergoing synchronous sporulation. (**c**) From our RNA-seq data, examples of a transcript that was spliced efficiently in spores—as indicated by a lack of reads mapping to the intronic sequence—and a transcript that showed, like Mat**a**1, intron retention. See Table 1 for a list of other transcripts and their intron retention in spores.

**Figure 4 biomolecules-11-01223-f004:**
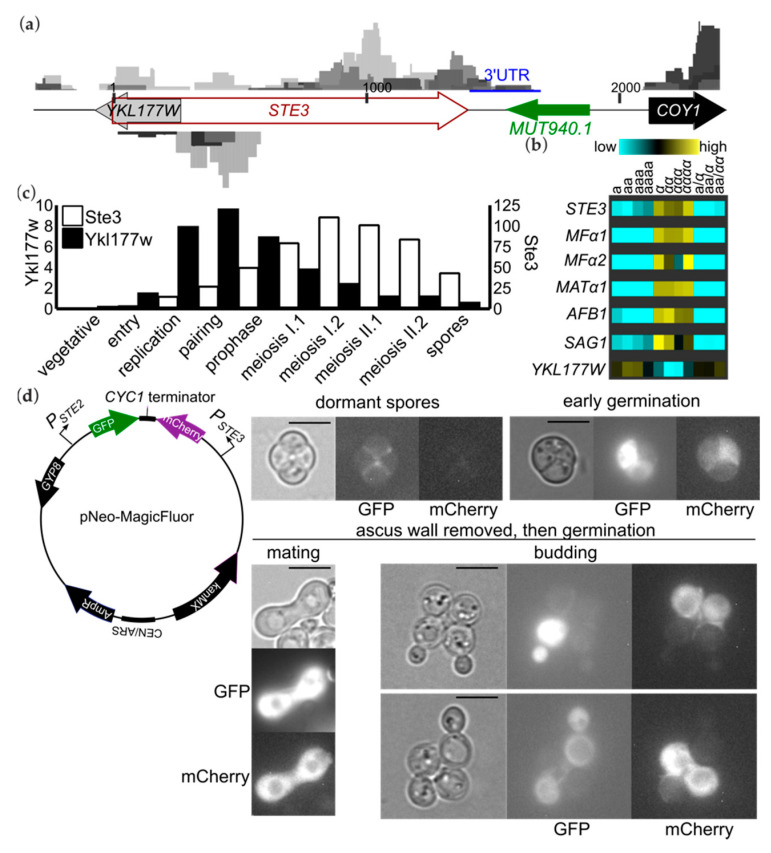
Ste3 and an antisense transcript, Ykl177w, co-exist in spores but the *STE2* and *STE3* promoters are sufficient for translational repression. (**a**) The structure of the *STE3* gene region is illustrated with ORFs drawn as large arrows. Numbers indicate basepairs, starting at the Ste3 start codon. RNA-seq reads from our four samples (two replicates each of **a** and α spores) were mapped to each strand and are shown as grayscale peaks, color-coded by sample. Those mapping to the strand representing Ste3 transcript are shown above, and those mapping to the other strand—presumably corresponding to Ykl177w—are shown below. The scales for plotting the peak heights are the same for all samples and both strands. *MUT940.1* indicates a non-coding RNA found in sporulating cells in another study [31] that partially overlaps the *STE3* 3′UTR. (**b**) Heatmap showing mating-type-dependent changes in gene expression based on microarray published data [32]. Source strains varied by mating type and ploidy, as indicated (e.g., “**aaaa**” is a tetraploid with the **a** allele at all four copies of *MAT*). (**c**) As in Figure 1a, RNA-seq data from [6] for Ste3 and Ykl177w. (**d**) The pNeo-MagicFluor plasmid carries the promoter sequences from *STE2* and *STE3* driving expression of GFP and mCherry, respectively. As a low-copy centromeric plasmid, it segregates randomly during sporulation and not all spores inherit it. Micrographs show transmitted light or GFP or mCherry fluorescence in mature asci in sporulation medium (“dormant spores”), following 2 h of exposure to rich medium (“early germination”), or following enzymatic removal of the ascus wall and 5 h of exposure to rich medium (“ascus wall removed, then germination”). Representative micrographs show spores undergoing mating or budding, as indicated. The pictured zygote was presumably formed from two spores from different asci. The budding spores are still attached to their meiotic sister spores by interspore bridges. Scale bars, 6 µm.

**Figure 5 biomolecules-11-01223-f005:**
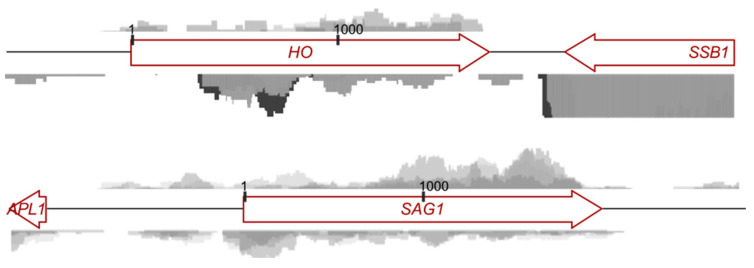
Antisense transcripts corresponding to *HO* or *SAG1* co-exist in spores. As in Figure 4a, RNA-seq reads from sorted spores were mapped to regions of the genome surrounding the *HO* or *SAG1* gene. Note that for the *HO* gene the scale was set the same for the sense and antisense reads mapping to *HO*, but at this scale most reads from the adjacent, highly-expressed *SSB1* gene exceed this scale.

**Table 1 biomolecules-11-01223-t001:** Intron retention in spore transcripts. RNA-seq reads from sorted spores were analyzed for 90 intron-containing genes to classify them as to the efficiency of their splicing in spores. The number of genes in each splicing efficiency category is given in parenthesis.

Genes with Retained Introns (49)	Genes with No Obvious Intron Retention ^1^ (23)	Ambiguous Genes (18)
*AMA1, APE2, ARP2, COF1, COX5B, DBP2, DCN1, DYN2, ECM33, ERV1, GCR1, GPI15, HNT1, HNT2, HOP2, LSM2, MCM21, MND1, MOB1, MRK1, MRPL44, MUD1, NSP1, PCH2, PFY1, PMI40, POP8, QCR10, QCR9, REC107, RPL17A, RPL22B, RPS17A, RPS18B, RUB1, SCS22, SMD2, STO1, TAD3, UBC13, UBC5, UBC9, VMA10, VMA9, YML6, YOP1, YRA1, YSC84, YSF3*	*ACT1*, *BIG1, BUD25*, *COX4*^2^*DMC1*, *GIM5*, *GLC7*, *LSB3*, *LSM7*, *MMS2*, *MOB2, MPT5*, *PHO85, PRE3*, *RAD14*, *RFA2*, *RIM1, SAE3*, *SRC1, TAN1*, *TUB1*, *UBC4*, *UBC8* ^1^	*ARP9*, *CIN2*, *CPT1*, *EPT1*, *HFM1*, *IMD4*, *MEI4*, *MTR2*^2^, *NMD2, REC102*, *REC114*, *SPO1*, *SPO22*, *SPT14*, *TUB3, UBC12*, *VPS75*, *YBR220C*

^1^ Includes genes with poor coverage where intron retention would likely go unnoticed. ^2^ Introns are outside the coding sequence.

## Data Availability

Most data are contained within the article or Appendix A. The RNA-seq data obtained in this study are openly available in the GEO database (accession number GSE180221). RNA-seq and Ribo-seq publicly available dataset GEO GSE34082 was also analyzed in this study. Microarray data from Appendix A of two published studies were taken either from the study directly [27] or accessed and plotted via the SPELL search engine (version 2.0.3r71) [32,64] running on the Saccharomyces Genome Database (http://yeastgenome.org, accessed on 16 August 2020).

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
