# Peer review of "Post-Transcriptional Control of Mating-Type Gene Expression during Gametogenesis in Saccharomyces cerevisiae"

_biomolecules, 2021, doi:10.3390/biom11081223_

Round 1

Reviewer 1 Report

The manuscript describes the results of transcript levels, splicing, and translational rates of mating-type specific genes during gametogenesis. The main result is related to the fact that Mata1 transcripts were mostly unspliced and that antisense transcripts to many mating genes were identified. However, no functional data linking these different pieces of data were presented.

Comments: 

- Lines 257-258: Munding et al., 2013 actually says: "Meiosis in yeast is triggered in part by nutrient signaling (Mitchell, 1994; Neiman, 2011), which also leads to transcriptional repression of RPGs (Chu et al., 1998; Gasch et al., 2000; Munding et al., 2010; Primig et al., 2000; Warner, 1999)." and "After their repression early in meiosis, RPGs are induced in late meiosis (Chu et al., 1998; Munding et al., 2010; Primig et al., 2000), even though the starvation conditions continue."  Therefore, the authors should clarify whether they are specifying early or late meiosis here.

- Lines 263-264: This information does not seem to be obviously stated in the cited reference, please clarify.

- Lines 348-349: It is not clear to this reviewer where this is presented in the Discussion section.

- Lines 356-357: What is the relative distribution of Ndt80 binding sites in the genome of S. cerevisiae? This information is important to understand what are the chances of any RNA transcriptional product be present close to an Ndt80 binding site. 

- Lines 346-347 and 363-364: Although different additional experiments could be performed to test this model, simply expressing an RNA corresponding to the portion of Ste3 gene that would be complementary, and therefore annealed by the "Ste3 antisense" should outcompete the antisense and give a functional insight to the data presented herein.

Author Response

- Lines 257-258: Munding et al., 2013 actually says: "Meiosis in yeast is triggered in part by nutrient signaling (Mitchell, 1994; Neiman, 2011), which also leads to transcriptional repression of RPGs (Chu et al., 1998; Gasch et al., 2000; Munding et al., 2010; Primig et al., 2000; Warner, 1999)." and "After their repression early in meiosis, RPGs are induced in late meiosis (Chu et al., 1998; Munding et al., 2010; Primig et al., 2000), even though the starvation conditions continue."  Therefore, the authors should clarify whether they are specifying early or late meiosis here.

This is an excellent point; we should have been clearer. We changed “During sporulation” to “Late in sporulation”.

- Lines 263-264: This information does not seem to be obviously stated in the cited reference, please clarify.

The relevant text in lines 263-264 reads: “It has been proposed that the very short sizes of the Mata1 introns (52 and 54 nucleotides) leads to inefficient splicing even in wild-type vegetative cells [29]”. The cited reference 29 includes the sentence “ This relative inefficient splicing might be caused by the small size of its introns (50 nt), which are smaller than most S. cerevisiae introns.” It is this information that we were referring to.

- Lines 348-349: It is not clear to this reviewer where this is presented in the Discussion section.

The relevant text in lines 348-349 reads: “If these transcripts anneal to each other, the resulting dsRNA may inhibit Ste3 translation (see Discussion).” In the Discussion section (lines 401-403) we wrote: “ribosomes stall at the site of overlap [40]. We detected complementary RNAs in the same cells which, if they form dsRNA, could interfere with translation.” The term “dsRNA” only appears in these two sections of the text.

- Lines 356-357: What is the relative distribution of Ndt80 binding sites in the genome of S. cerevisiae? This information is important to understand what are the chances of any RNA transcriptional product be present close to an Ndt80 binding site.

Reviewer #2 made a similar point about the relevance of the putative MSE sites. When we searched the entire genome in an unbiased manner, we found numerous putative Ndt80 binding sites/MSEs, to the extent that those we noted in our genes of interest are likely not enriched above expectations from random distributions. We also checked for evolutionary conservation of a few and they didn’t seem particularly conserved. Hence we have removed references to putative Ndt80 sites/MSEs from the text and figures.

- Lines 346-347 and 363-364: Although different additional experiments could be performed to test this model, simply expressing an RNA corresponding to the portion of Ste3 gene that would be complementary, and therefore annealed by the "Ste3 antisense" should outcompete the antisense and give a functional insight to the data presented herein.

We thank the reviewer for this suggestion. However, the experiment is not as simple as it seems. Since codon 53 in the Ste3 ORF is also an AUG, it may be technically difficult to produce a sense transcript complementary to the antisense that does NOT get translated into a protein; a truncated Ste3 protein could act in a dominant-negative manner to interfere with pheromone signaling and obscure any functional output. Furthermore, we are not confident that our RNA-seq data precisely identify the 5’ and 3’ ends of the antisense transcript, which would require additional experiments. We would expect, if our model is correct, a decrease in mating upon spore germination as a result of receptor inhibition. Currently the only published assay – time-lapse microscopy of germinating asci to distinguish morphologically budding from mating – is both laborious and often (in our hands) ambiguous. We have engineered two new colony-growth-based assays to more quantitatively distinguish spore mating from budding, but these are not yet published, and require some additional refinement. Finally, the major obstacle is the apparent sufficiency of the STE3 promoter to repress translation in spores, independent of the antisense transcript. Unless we ablate this mechanism, the details of which we still do not understand, we would likely miss any functional effect of the reviewer’s suggested experiment. We realized that this point – that the “promoter-autonomous” translational repression is likely redundant with the putative antisense repression – was not clearly stated in the text, so we have amended the Discussion to clarify.

Reviewer 2 Report

This paper used a nice strategy to analyze the transcriptomes of isolated spores of the two different mating types in S. cerevisiae. Surprisingly, they found only one differentially expressed transcript (apart from the control investigator-inserted transcripts used to allow them to distinguish the mating types of the spores). For the other transcripts that are known to be mating-type-specific in vegetative cells, they found that the transcripts were present in both spore mating types. Mining a variety of data from other studies, as well as their new experiments, led to several further conclusions. Of greatest interest: (i) several mating-type-specific gene transcripts are translationally inactive in spores; (ii) promoters of some mating-type-specific genes (STE2, STE3, MFalpha1) suffice to yield spore-specific translational repression; (iii) spores contain antisense transcripts for many mating-type-specific genes; and (iv) the one differentially expressed transcript (MATa1) is incompletely spliced. The aggregate picture is that spores do not undergo any mating-type-specific gene expression until they germinate and become vegetative haploid cells.

The analysis is thoughtful, interesting, and knowledgeable about the literature, typical of McMurray lab papers. It is a worthy contribution and I have no major issues with the paper.

Minor comments:

  1. Heading 3.1 mentions two transcripts that differ between spore mating types, but one of those (YAR064W) was a false positive, so better to say ONE transcript.
  2. Lines 267-269: intron lengths listed in reverse order?
  3. The ime4 data is cited as being in Fig. S3 (line 441) but is actually part of Fig. 3A. It was confusing to see the data long before it was mentioned in the text, and I would suggest removing these data as they do not add to the paper. Similarly, I would remove the discussion of m6A as it seems irrelevant to the findings of the paper.
  4. The MSE sites and associated discussion of Ndt80 seemed distracting and without a corroborating experiment I would recommend they be eliminated.
  5. The discussion of “transperons” (lines 446-459) seemed pointless as they do not appear to address the findings of the paper.
  6. The discussion of Asg7 was more detailed than is needed to make the main point (that there exist mechanisms to prevent inappropriate behaviors should genes from both mating types be expressed in the same spore) and could be shortened.

Author Response

1. Heading 3.1 mentions two transcripts that differ between spore mating types, but one of those (YAR064W) was a false positive, so better to say ONE transcript.

This is an excellent point and we made this change.

2. Lines 267-269: intron lengths listed in reverse order?

The intron lengths were listed in the correct order, but we noticed a typo that may have caused some confusion: we replaced “that” (capitalized in the following sentence) with “than”: “the minimum intron length was 56 or 58 nucleotides in the genes with or without intron retention, and the median and mean intron lengths were greater in the genes with retained introns THAT in those without (99 vs 96 and 184.8 vs 173.1, respectively).”

3. The ime4 data is cited as being in Fig. S3 (line 441) but is actually part of Fig. 3A. It was confusing to see the data long before it was mentioned in the text, and I would suggest removing these data as they do not add to the paper. Similarly, I would remove the discussion of m6A as it seems irrelevant to the findings of the paper.

We had somehow overlooked the fact that the raw ime4 data were still in Fig. 3A. We agree that this is confusing and have removed those data and the discussion of them. However, since (i) m6A is only found during sporulation in yeast; (ii) at least two transcripts of interest are known to be m6A-modified (Ste3 and Ste6), and (iii) in other contexts m6A modification can inhibit processing or translation, we kept the text stating these points and that Mata1 does not appear to be modified by m6A.

4. The MSE sites and associated discussion of Ndt80 seemed distracting and without a corroborating experiment I would recommend they be eliminated.

Reviewer #1 made a similar point about the relevance of the putative MSE sites. When we searched the entire genome in an unbiased manner, we found numerous putative Ndt80 binding sites/MSEs, to the extent that those we noted in our genes of interest are likely not enriched above expectations from random distributions. We also checked for evolutionary conservation of a few and they didn’t seem particularly conserved. Hence we have removed references to putative Ndt80 sites/MSEs from the text and figures.

5. The discussion of “transperons” (lines 446-459) seemed pointless as they do not appear to address the findings of the paper.

We corrected the first sentence of this paragraph to clarify why we think “transperons” are worth mentioning (added text is capitalized): “RNA-binding proteins bind and control the translational timing of various mRNAs that are transcribed early DURING SPORULATION but are not translated until later.” There is such ample precedent for RNA-binding proteins repressing the translation of mRNAs during sporulation that we think we would be remiss if we failed to mention that RNA-binding proteins were recently found to bind together and control the translation of many of the very same genes we analyzed in our study. Furthermore, given our findings suggesting that the STE2 and STE3 promoters are sufficient for repression, we think it is worth noting that the sequences driving transperon assembly fall in the coding regions.

6. The discussion of Asg7 was more detailed than is needed to make the main point (that there exist mechanisms to prevent inappropriate behaviors should genes from both mating types be expressed in the same spore) and could be shortened.

We thank the reviewer for this point and have shortened the text accordingly.